# Transdifferentiation mediated tumor suppression by the endoplasmic reticulum stress sensor IRE-1 in *C. elegans*

Mor Levi-Ferber, Hai Gian, Reut Dudkevich, Sivan Henis-Korenblit*

The Mina and Everard Goodman Faculty of Life Sciences, Bar-Ilan University, Ramat-Gan, Israel

**Abstract** Deciphering effective ways to suppress tumor progression and to overcome acquired apoptosis resistance of tumor cells are major challenges in the tumor therapy field. We propose a new concept by which tumor progression can be suppressed by manipulating tumor cell identity. In this study, we examined the effect of ER stress on apoptosis resistant tumorous cells in a *Caenorhabditis elegans* germline tumor model. We discovered that ER stress suppressed the progression of the lethal germline tumor by activating the ER stress sensor IRE-1. This suppression was associated with the induction of germ cell transdifferentiation into ectopic somatic cells. Strikingly, transdifferentiation of the tumorous germ cells restored their ability to execute apoptosis and enabled their subsequent removal from the gonad. Our results indicate that tumor cell transdifferentiation has the potential to combat cancer and overcome the escape of tumor cells from the cell death machinery.

*For correspondence: sivan.korenblit@biu.ac.il

Competing interests: The authors declare that no competing interests exist.

## Introduction

A major challenge in the tumor therapy field is the development of new strategies to eliminate tumors and cancer cells. Whereas most of the current therapeutic strategies are based on apoptosis induction in the tumor cells, the effectiveness of these approaches is limited due to acquired apoptosis resistance (*Hanahan and Weinberg, 2000, 2011*). Thus, deciphering ways to restore apoptosis sensitivity to tumorous cells that acquired apoptosis resistance may revive 'old' tools with therapeutic potential to eliminate tumor cells.

The *gld-1* (GermLine Development defective) gene encodes a germline-specific QUAKING-like RNA binding protein, which represses the translation of a variety of germline transcripts (*Jungkamp et al., 2011; Wright et al., 2011*). Consequently, GLD-1 regulates many aspects of germ cell biology (*Francis et al., 1995a, 1995b; Kadyk and Kimble, 1998; Jan et al., 1999; Hansen et al., 2004; Ciosk et al., 2006*). One of the striking consequence of a deficiency in *gld-1* is the formation of a proximal germline tumor that fills the gonad (*Francis et al., 1995a*). This germline tumor is the result of re-entry of meiotic germ cells into the mitotic cell cycle instead of maturing into oocytes (*Francis et al., 1995a*). Importantly, some aspects of tumorigenesis are exhibited in the *gld-1* germline tumor model. These include the ability of the tumorous germ cells to proliferate in a growth factor–independent manner (*Francis et al., 1995a*) and their regulation by genes homologous to known human oncogenes or human tumor suppressor genes (*Pinkston-Gosse and Kenyon, 2007*). Notably, these tumorous germ cells acquired resistance to apoptosis (*Gumienny et al., 1999*). In addition, some precocious germ cell transdifferentiation into ectopic somatic cells has been reported to occur at a low frequency in *gld-1*-deficient animal (*Ciosk et al., 2006*). This transdifferentiation of the germ cells can be further enhanced by manipulation of RNA binding proteins, P granule components, transcription factors and histone modifiers; all of which regulate gene expression within the transdifferentiating germ

**eLife digest** If a cell in the body becomes damaged or stops working properly, it can trigger its own destruction. This helps to prevent the accumulation of damaged cells. However, cancer cells can often tolerate much greater damage than normal cells. Toxic chemotherapies, which are often used to treat cancer, work by severely damaging the cells to help trigger their self-destruction. Unfortunately, chemotherapy does not work on all cancer cells, and the remaining treatment-resistant cells may continue to grow and spread in more aggressive ways.

Now, Levi-Ferber et al. have found a way to change the identity of cancer cells, which makes them more likely to self-destruct. The experiments used roundworms called *Caenorhabditis elegans* that had a genetic mutation that causes them to develop tumors in their reproductive organs. Normally, the cells in these tumors do not self-destruct. Levi-Ferber et al. exposed tumor cells from the worms to chemicals or to genetic modifications that cause unfolded proteins to accumulate inside the cell. This build-up of proteins stresses a structure in the cell called the endoplasmic reticulum. Normally, if endoplasmic reticulum stress gets too high, the cell activates various pathways to relieve the stress, and if these fail, the cell self-destructs.

Levi-Ferber et al. showed that a protein called IRE-1, which senses endoplasmic reticulum stress, caused the tumor cells to change into a type of non-cancerous cell. After the change, the cells were also more sensitive to self-destruction. This meant that tumors grew more slowly and ended up smaller, allowing the animals to survive longer.

Together, the experiments suggest that treatments that force cancer cells to become a different cell type might be one way to prevent the emergence of treatment-resistant tumor cells. Future research will be needed to investigate exactly how IRE-1 causes the identity of the cell to change, and to see if this process could treat other kinds of cancer.

cells themselves (*Ciosk et al., 2006*; *Tursun et al., 2011*; *Patel et al., 2012*; *Updike et al., 2014*). In this study, we investigated whether cellular and organismal stress can affect germ cell fate and tumorigenicity in the *gld-1* tumor model.

## Results

### ER stress induces apoptosis in the gonads of *gld-1*-deficient animals

In *Caenorhabditis elegans*, chemically or genetically-induced ER stress promote germ cell apoptosis in normal (i.e., non-tumorous) germlines (*Levi-Ferber et al., 2014*). Nevertheless, the germ cells in *gld-1*-deficient animals were reported to be resistant to physiological and DNA damage-induced apoptosis (*Gumienny et al., 1999*), suggesting that they may have acquired global resistance to apoptosis promoting signals, a common phenomenon of transformed cells. To directly test whether ER stress succeeds or fails to induce apoptosis in *gld-1*-deficient animals, we exposed the animals to ER stress, and assessed the presence of apoptotic corpses within the gonad using the SYTO12 apoptotic dye. ER stress was induced either by treating the animals with tunicamycin (a specific inhibitor of N-glycosylation) or by treating the animals with *tfg-1* RNAi (*tfg-1* encodes a component of COPII-coated vesicles required for the export of cargo from the ER [*Witte et al., 2011*]). Both treatments specifically induce ER stress (*Levi-Ferber et al., 2014*). As previously reported (*Gumienny et al., 1999*), no apoptotic corpses representing physiological germ cell apoptosis were detected in the tumorous gonads in the absence of ER stress (*Figure 1A,B* and *Figure 1—figure supplement 1*). However, we consistently detected SYTO12-labeled corpses in tumorous gonads of *gld-1* RNAi-treated animals exposed to ER stress induced either by genetic means (i.e., *tfg-1* RNAi) or by chemical means (i.e., tunicamycin) (*Figure 1A,B* and *Figure 1—figure supplement 1*).

We further confirmed the presence of engulfed apoptotic corpses in the ER stressed tumorous gonads using CED-1::GFP expressed in the engulfing pseudopodia of the gonadal sheath cells (*Figure 1D*). CED-1::GFP-engulfed apoptotic corpses were only detected in tumorous gonads upon induction of ER stress. These differences were not the result of restoration of *gld-1* expression in the stressed animals as GLD-1 protein levels were similarly reduced in animals treated with *gld-1* RNAi along with control RNAi or *tfg-1* RNAi (*Figure 1—figure supplement 2*).

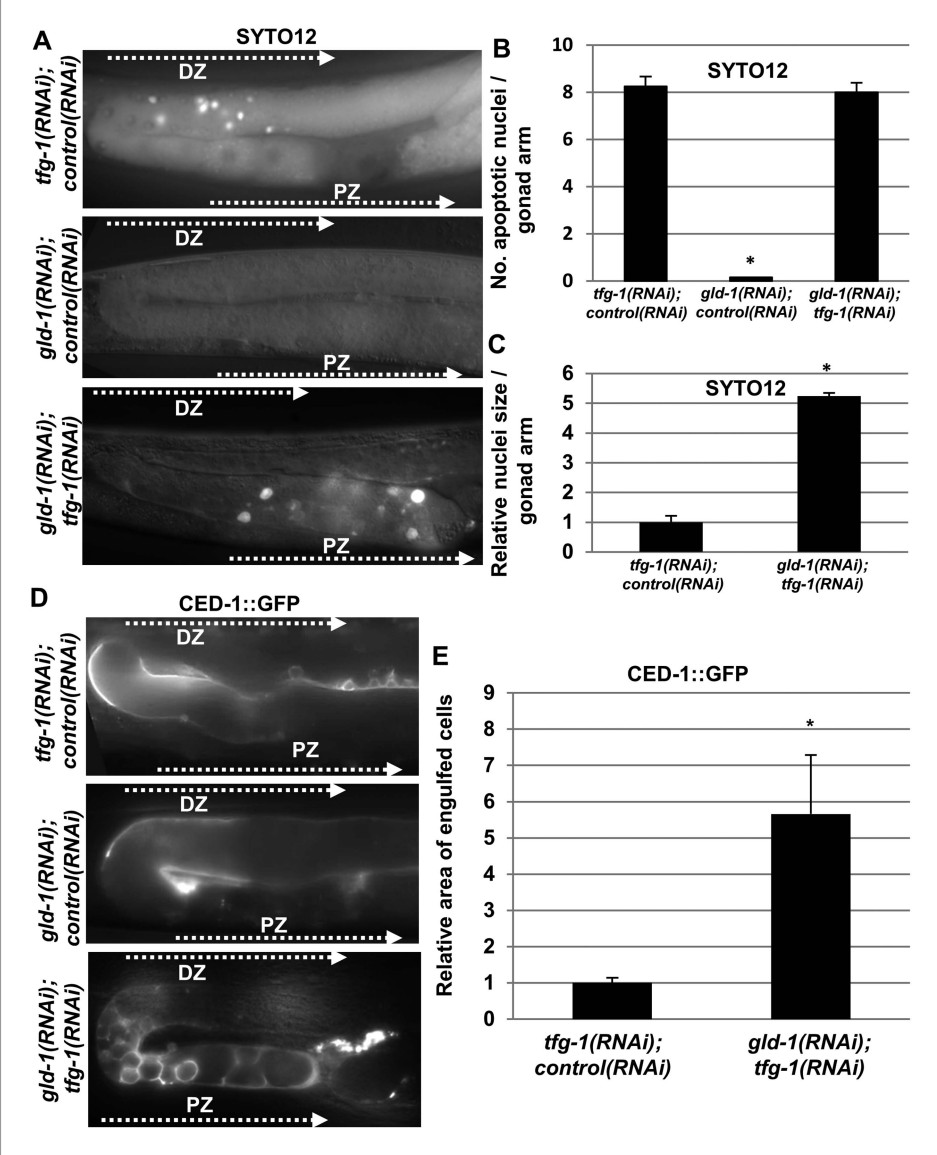

**Figure 1**. Apoptotic corpses are detected in the gonads of *gld-1*-deficient animals upon induction of ER stress. (**A–C**) Day-3 animals treated with the indicated RNAi were stained with SYTO12 to detect apoptotic cell corpses. The average number of SYTO12-labeled apoptotic corpses per gonad is shown in **B**. The relative size of the SYTO12-labeled nuclei is shown in **C**. See *Figure 1—figure supplement 1* for SYTO12-labeling of tunicamycin treated animals. (**D**, **E**) CED-1::GFP expressed in the gonadal sheath cells was used to follow engulfment of apoptotic cells within the gonad of day-3 animals. The relative average area of the engulfed cells is shown in **E**. Note that in non-tumorous animals the apoptotic cells are detected in the distal gonad zone (DZ), whereas in the ER stressed-tumorous animals they are detected in the proximal gonad zone (PZ). Asterisk marks Student's T-test values of $p < 0.001$ compared to animals treated with a mixture of control and *tfg-1* RNAi. *gld-1* RNAi knocked down GLD-1 protein levels to a similar extent upon treatment with control or *tfg-1* RNAi (see *Figure 1—figure supplement 2*). At least 40 gonads of each genotype were analyzed.

The following figure supplements are available for figure 1:

**Figure supplement 1**. Apoptotic cell corpses are detected in the gonads of tunicamycin-treated tumorous animals.

**Figure supplement 2**. GLD-1 protein levels are efficiently reduced by *gld-1* RNAi in the single, double and triple RNAi mixtures.

## The apoptotic corpses in the gonad of *gld-1*-deficient animals differ from typical germ cell corpses

Remarkably, the apoptotic corpses in the tumorous gonads of ER-stressed animals were distinct from those detected in non-tumorous gonads in several ways. First, in terms of location within the gonad—whereas in non-tumorous gonads apoptotic germ cell corpses were usually detected at the turn of the gonad, where oogenesis occurs, those detected in ER-stressed tumorous gonads were located in the proximal region of the gonad (*Figure 1A,D*). In addition, in terms of size, both the SYTO12-labeled nuclei and the CED-1::GFP-engulfed cells in the tumorous gonads exposed to ER stress were significantly larger than those detected in non-tumorous animals (*Figure 1A,C,D,E*). Furthermore, unlike engulfed germ cell corpses that are typically round, the CED-1::GFP engulfed cells in the ER-stressed tumorous gonads displayed a variety of shapes (*Figure 1D*). Altogether, the differences in size, shape and location suggest that the apoptotic cell corpses in ER-stressed tumorous gonads are distinct from the germ cell corpses observed in non-tumorous gonads.

## Ectopic cells with large nuclei accumulate in the gonads of *gld-1; ced-3*-deficient animals

We hypothesized that if cells undergoing apoptosis were continuously engulfed and removed from the stressed tumorous gonads, then blocking apoptosis should result in their accumulation in the gonads. To this end, *ced-3* expression was inactivated to prevent apoptosis and the pattern of the nuclei in the gonad was assessed by DAPI (4′,6-diamidino-2-phenylindole) staining. Strikingly, blocking apoptosis resulted in the accumulation of cells with large and misshaped nuclei occupying nearly 40% of the stressed gonad (*Figure 2A,B*). Even without exposing the animals to ER stress, blocking apoptosis resulted in the accumulation of ectopic cells with large and misshaped nuclei in the gonads of the *gld-1* RNAi-treated animals. However, in the absence of ER stress, the abnormally large nuclei occupied only 10% of the gonad (*Figure 2A,B*), indicating that ER stress promotes their induction. Interestingly, blocking apoptosis enabled the detection of ectopic cells within the gonads of nearly 90% of the *gld-1* RNAi-treated animals examined, both under normal growth conditions as well as under ER stress, albeit to different extents (*Figure 2—figure supplement 1* and *Figure 2A,B*). The enhanced accumulation of ectopic cells in the ER stressed gonads was not due to altered efficiency of the *gld-1* RNAi as the GLD-1 protein levels were similarly reduced upon *gld-1* RNAi treatment in combination with control/*tfg-1* and *ced-3* RNAi (*Figure 1—figure supplement 2*). Furthermore, treatment of *gld-1(q485); ced-3(n1286)* double mutants with ER-stress inducing *tfg-1* RNAi also increased the accumulation of cells with large nuclei in the gonads (*Figure 2C*), recapitulating the phenomena observed in *gld-1, tfg-1* and *ced-3* RNAi-treated animals. Altogether, these findings suggest that these cells with large nuclei in the gonads of *gld-1*-deficient animals are the ones that are normally cleared from the gonad by apoptosis, and not the typical germ cells.

## The ectopic cells in the gonad of *gld-1*-deficient animals express somatic markers

A low frequency of precocious activation of embryonic-like differentiation of the germ cells in *gld-1*-deficient animals has been reported before (*Ciosk et al., 2006*). Hence, we suspected that the abnormal nuclei within the gonad of *gld-1*-deficient animals, which were further induced upon ER stress, could be differentiated somatic cells as well. Thus, we examined the expression of transgenic somatic markers within the gonad of apoptosis-defective animals. In accordance with the detection of ectopic cells in the tumorous gonads upon blockage of apoptosis, *ced-3* inactivation allowed the detection of differentiation markers in 80–90% of the gonads of *gld-1*-RNAi treated animals (*Figure 2—figure supplement 1*). Genetically-induced ER stress resulted in a 3–6 fold increase in the fluorescence of each of the three germ layers transgenic markers in the tumorous gonads (*Figure 3A*). Similarly, treatment with tunicamycin increased the fluorescence of the neuronal marker *Punc-119::gfp* in the tumorous gonads of *gld-1; ced-3* deficient animals (*Figure 3B*).

Do the cells with abnormal large nuclei also express the somatic markers, or are these two separate abnormal cell populations that arise in the tumorous gonad upon ER stress? Analysis of the fluorescence pattern of a *Pelt-2::NLS::gfp* intestinal somatic marker, which by virtue of its NLS signal specifically labeled the nuclei of the corresponding somatic cells, revealed that these nuclei were larger than those of typical germ cells (*Figure 3A*). In addition, Hoechst-nuclei staining of a strain

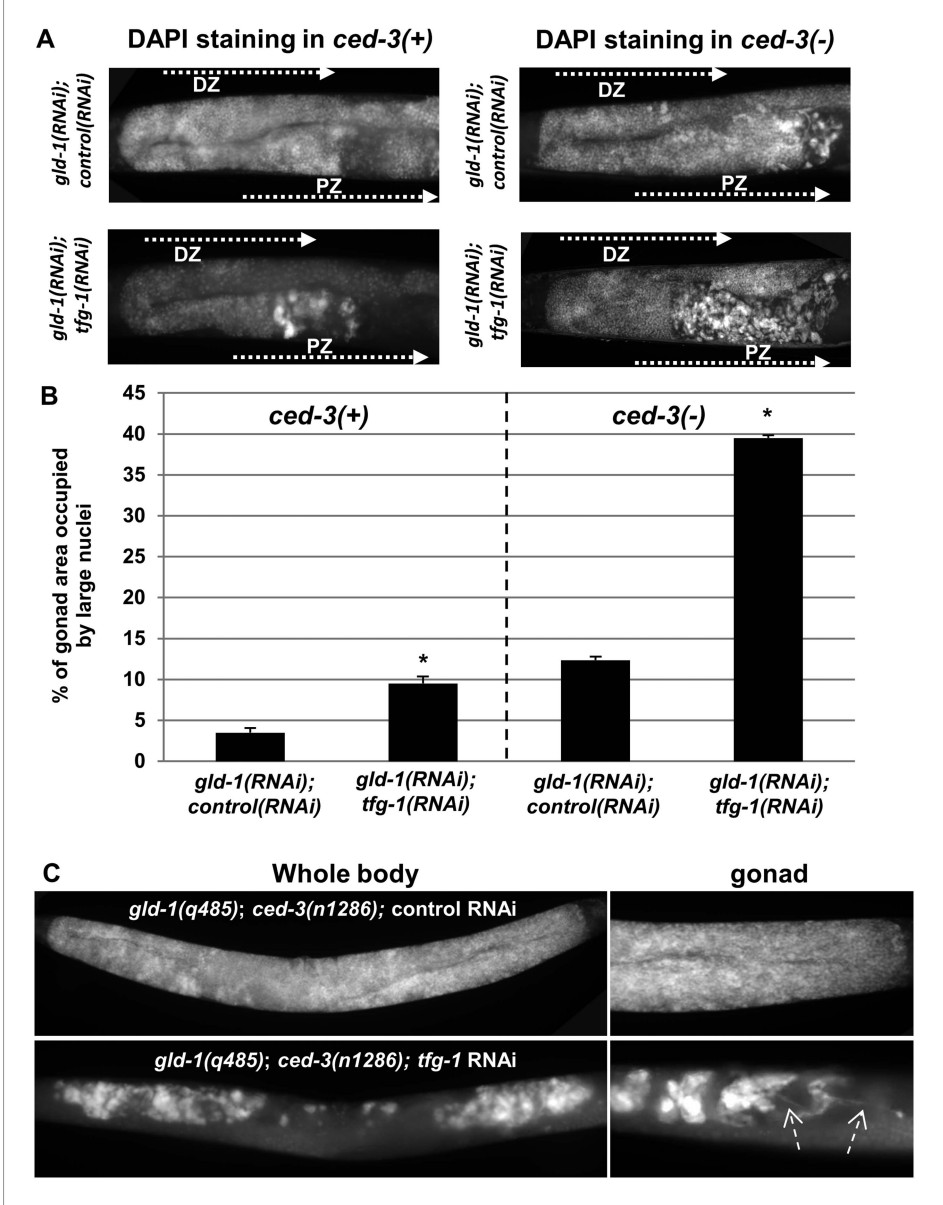

**Figure 2**. Ectopic cells with large nuclei accumulate in the gonads of *gld-1; ced-3* animals. (**A**) Representative micrographs (x400) of DAPI-stained gonads of day-4 animals. Animals were treated with the indicated RNAi. *gld-1* RNAi was used to induce a germline tumor. *ced-3* RNAi served to block apoptosis. *tfg-1* RNAi was used to induce ER stress. Treatment with *tfg-1* RNAi increased the levels of ectopic cells with large misshaped nuclei at the proximal zone of the gonad of *gld-1* deficient animals, especially upon apoptosis inactivation. DZ marks the distal zone of the gonad. PZ marks the proximal zone of gonad. (**B**) Bar graph presents percentage of gonad area occupied by large nuclei in the indicated genotypes (n = at least 60 gonads per genotype). Asterisks mark Student's T-test values of p < 0.001 of *tfg-1* RNAi-treated animals compared to their non-stressed controls. Note that ectopic cells with large nuclei were detected to different extents in most of the animals examined (see *Figure 2—figure supplement 1*). (**C**) The induction of ectopic cells in the gonad by *tfg-1*-induced ER stress was recapitulated in *gld-1(q485); ced-3 (n1286)* double mutants. Arrows point at axon-like structures detected within the gonads of *gld-1*-deficient animals upon ER stress.

The following figure supplement is available for figure 2:

**Figure supplement 1**. Ectopic somatic cells are detected in most of the gonads of *gld-1*-deficient animals.

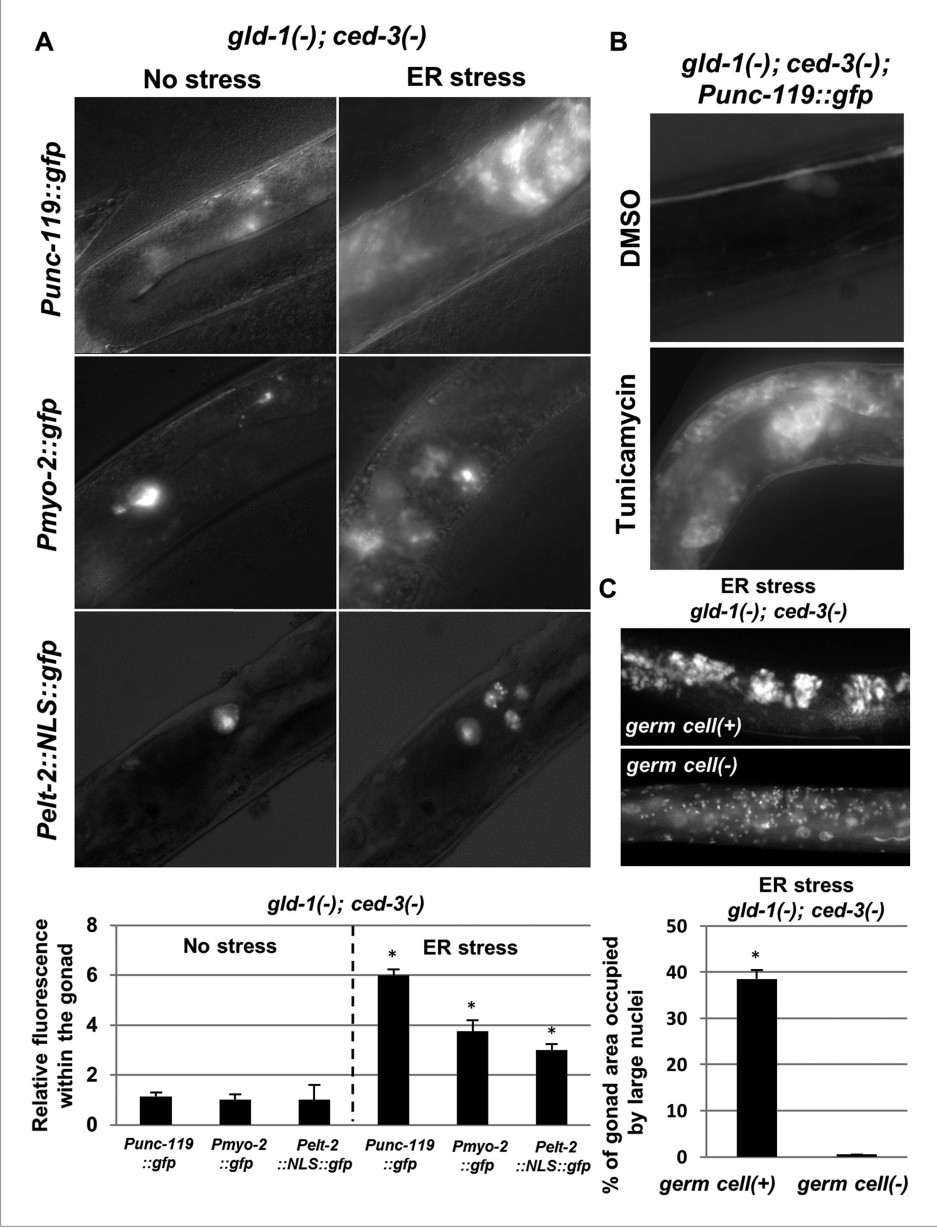

**Figure 3**. The ectopic somatic cells in the ER-stressed gonad of *gld-1*-deficient animals are germ cell-derived differentiated somatic cells. (**A**) Representative fluorescence micrographs (x400) of somatic differentiation markers expressed in the gonads of *gld-1; ced-3*-deficient animals on day-4 of adulthood. *gld-1* RNAi was used to sensitize the germline for transdifferentiation. *ced-3* RNAi was used to prevent the clearance of the ectopic somatic cells from the gonad. *Punc-119::gfp* is a neuronal marker, *Pmyo-2::gfp* is a pharyngeal muscle marker. *Pelt-2::NLS::gfp* is an intestinal marker. ER stress was induced by *tfg-1* RNAi or by an *xbp-1* mutation. Asterisks mark Student's T-test values of p < 0.001 compared to non-stressed conditions. At least 40 animals were analyzed per genotype. (**B**) Representative fluorescence micrographs (x400) of *Punc-119::gfp* in the gonads of *ced-3(n1286)* day-4 animals treated with *gld-1* RNAi. ER stress was induced chemically with tunicamycin and compared to DMSO treatment. (**C**) Representative fluorescence micrographs of DAPI-stained nuclei of ER-stressed day 4 adults treated with a mixture of *gld-1, ced-3* and *tfg-1* RNAi. Accumulation of abnormal somatic-like nuclei was detected in germ cell(+) animals and not in germ cell(–) *glp-1(–)* mutants. At least 50 gonads were analyzed per genotype. Asterisk marks Student's T-test values of p < 0.001. See *Figure 3—figure supplement 1* for co-localization of the somatic marker expressing cells and the cells with the large nuclei and/or the cell under engulfment.

The following figure supplement is available for figure 3:

**Figure supplement 1**. The ectopic somatic cells in the tumorous gonad have large nuclei and are engulfed by the surrounding gonad.

expressing the neuronal *Punc-119::gfp* somatic marker was used to assess whether the pattern of the ectopic somatic cells in the gonad overlapped with the cells harboring small germline-like nuclei or whether it overlapped with the ectopically large nuclei. We found that under ER stress conditions, the cells that expressed the neuronal marker co-localized with the cells in the gonad that had atypically large nuclei. No expression of the *Punc-119::gfp* somatic marker was detected in regions of the gonad devoid of large-nucleated cells (*Figure 3—figure supplement 1A*).

In addition, we examined whether the cells that undergo apoptosis in the stressed tumorous gonad are the ones that express the somatic markers. To this end, we examined a strain that co-expresses *ced-1::gfp* in the sheath cells of the gonad as well as the intestinal marker *Pelt-2::gfp::NLS*. Indeed, some of the engulfed cells also expressed the somatic marker (*Figure 3—figure supplement 1B*). These findings are consistent with the interpretation that the ectopic somatic cells that arise in the stressed tumorous gonad normally undergo apoptosis and are consequently engulfed and cleared from the gonad.

Why was the expression of the intestinal somatic marker apparent only in some of the *ced-1::gfp* labeled phagosomes and not in all of them? Likewise, why weren't all the large-nucleated cells co-localized with the cells in the gonad that expressed the neuronal marker? These phenomena are likely the consequence of the fact that the teratoma contains somatic cells of the different germ-layers, implying that only a fraction of the teratoma cells can be detected using somatic markers of specific tissues or germ layers.

## The ectopic cells in the gonad of *gld-1*-deficient animals are germ cell-derived somatic cells

The somatic cells in *gld-1*-deficient animals arise by precocious differentiation and loss of pluripotency of the germ cells (*Ciosk et al., 2006*). Thus, we explored whether the ectopic somatic cells in the gonad of ER stressed-animals are derived from germ cells as well. To this end, we treated animals with a normal germline or *glp-1(lof)* mutants, whose primary germ cells do not proliferate at non permissive temperatures, with a mixture of *gld-1, tfg-1* and *ced-3* RNAi, in order to generate optimal conditions for the detection of ectopic somatic cells in the gonad (RNAi efficacy was confirmed as described in 'Materials and methods'). Day-4 adults were analyzed for the presence of DAPI-stained ectopic large nuclei. Ectopic somatic cells were not detected in the gonad of germ cell-deficient *glp-1* animals, although they were detected in control germ cell (+) animals upon induction of ER stress (*Figure 3C*). These findings are consistent with the possibility that the ectopic somatic cells induced by ER stress in the tumorous gonad are derived from the germ cells themselves, uncovering ER stress as a potent regulator of germ cell pluripotency. Importantly, unlike previously identified regulators of germ cell pluripotency, all of which act at the final steps of the transdifferentiation process by directly regulating gene expression (*Ciosk et al., 2006*; *Tursun et al., 2011*; *Patel et al., 2012*; *Updike et al., 2014*); ER stress and ER homeostasis likely act further upstream, linking between cellular and organismal physiology and germ cell fate.

## *ire-1* promotes ER stress-induced germ cell transdifferentiation in the gonad of *gld-1*-deficient animals independently of *xbp-1*

After establishing that ER stress promotes germ cell transdifferentiation in *gld-1*-deficient animals we examined whether this induction is mediated by one of the ER stress sensor proteins. To this end, wild-type animals and mutant animals deficient in one of the ER stress sensors *ire-1, pek-1* or *atf-6* were treated with a mixture of *gld-1, ced-3* and *tfg-1* RNAi. At day 4 of adulthood, the presence of ectopic cells in the gonad was assessed by staining of the animals' nuclei with DAPI. We detected significantly increased levels of ectopic nuclei in the gonads of wild-type, *pek-1*-deficient and *atf-6*-deficient animals, in the presence of *tfg-1* RNAi (*Figure 4A*). In contrast, *tfg-1* RNAi failed to increase germ cell transdifferentiation in *ire-1*-deficient animals (*Figure 4A*). Nevertheless, a low level of ectopic somatic cells, similar to that observed in *gld-1*-deficient animals that were not exposed to ER stress, was still detected in *gld-1; ire-1*-deficient animals (*Figure 4A*). This basal level of germ cell transdifferentiation was not increased by *tfg-1* RNAi-induced ER stress or by the ER stress associated with the *ire-1* deficiency. We conclude that *ire-1* is not required for germ cell transdifferentiation per se. However, it is required for germ cell transdifferentiation in response to ER stress. Furthermore, these findings indicate that the regulation of germ cell pluripotency by ER stress in *gld-1*-deficient animals requires the activation of the ER stress sensor IRE-1, and is not simply the result of interference with ER homeostasis and function.

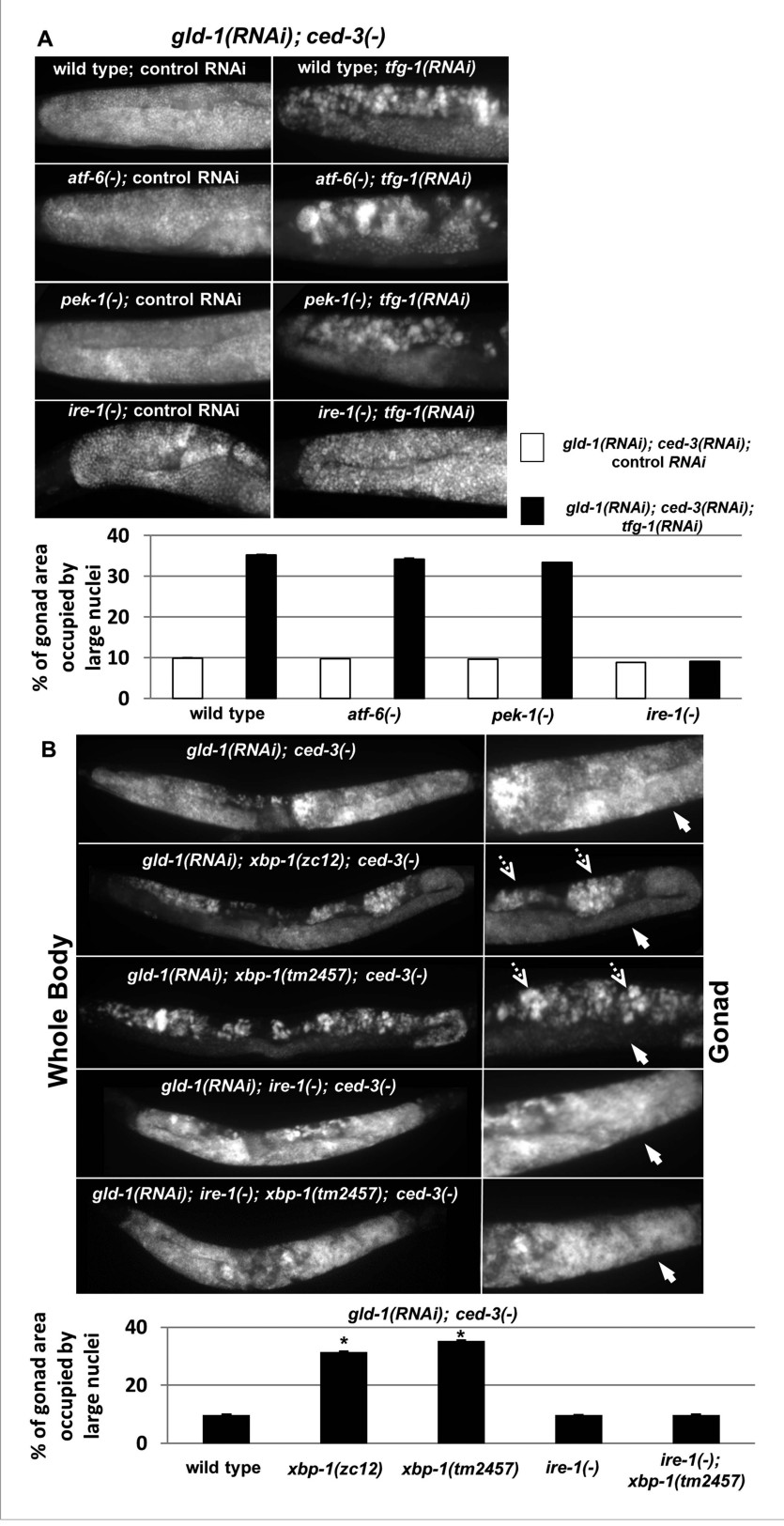

**Figure 4**. ER stress induces germline transdifferentiation in an *ire-1*-dependent but *xbp-1*-independent manner. (**A**) Representative micrographs (x400) of DAPI-stained gonads of day-4 animals treated with either a mixture of control, *gld-1* and *ced-3* RNAi or with a mixture of *tfg-1*, *gld-1* and *ced-3* RNAi. Treatment with *tfg-1*, *gld-1* and *ced-3* RNAi failed to induce germ cell transdifferentiation in *ire-1* mutants. (**B**) Representative micrographs of whole body
*Figure 4. continued on next page*

*Figure 4. Continued*

(x100) and gonads (x400) of DAPI-stained day-4 animals of the indicated genotypes treated with *gld-1* and *ced-3* RNAi. Solid arrows indicate mitotic germ cells. Dashed arrows indicate somatic nuclei. Bar graphs present percentage of gonad area occupied by ectopic cells. Asterisk marks Student's T-test of p < 0.001 relative to wild-type animals. Bar graphs present percentage of gonad area occupied by ectopic cells of the indicated genotypes (n = at least 70 gonads per genotype). Asterisks mark Student's T-test of p < 0.001 relative to the same animals treated with control, *gld-1*and *ced-3* RNAi. Note that both alleles of *xbp-1* similarly increased the percentage of gonad area occupied by ectopic somatic cells (p = 0.23).

Although one of the main downstream targets of *ire-1* is the ER stress-regulated transcription factor *xbp-1* (*Shen et al., 2001*; *Calfon et al., 2002*), *ire-1* can also directly activate signaling cascades by virtue of its oligomerization propensity (*Urano et al., 2000*; *Yoneda et al., 2001*), or regulate mRNA stability by virtue of its RNase activity via the RIDD pathway (*Hollien and Weissman, 2006*; *Han et al., 2009*). We hypothesized that if *ire-1* promotes germ cell transdifferentiation by activating the transcription factor *xbp-1*, then ER stress would fail to induce germ cell transdifferentiation in *xbp-1*-deficient animals. Initially, we used DAPI staining to assess the levels of germ cell transdifferentiation in two strains deficient in *xbp-1- xbp-1(zc12)* and *xbp-1(tm2457)*. Strikingly, we detected significantly increased levels of ectopic nuclei in the gonads of both *xbp-1* mutants upon treatment with a mixture of *gld-1* and *ced-3* RNAi on day 4 of adulthood (*Figure 4B*). Previous studies in mice and in *C. elegans xbp-1 (zc12)* mutants have demonstrated that the mere deficiency in XBP1 leads to activation of the ER stress sensor IRE1 (*Hu et al., 2009*; *Richardson et al., 2011*; *Hur et al., 2012*; *Safra et al., 2013*) Thus, we hypothesized that this high basal level of germ cell transdifferentiation in *xbp-1* mutants may be due to the increased activity of IRE-1 in these mutants. Consistent with this interpretation, the high level of germ cell transdifferentiation in *xbp-1* mutants was completely dependent on *ire-1* (*Figure 4B*). Thus, although ER stress-induced germ cell transdifferentiation is completely dependent on the ER stress sensor gene *ire-1*, it is not dependent on its downstream target *xbp-1*, implying that it is mediated by an *ire-1*-dependent *xbp-1*-independent signal.

## ER stress suppresses the germline tumor in an *ire-1* dependent manner

Since we have found that ER stress-induced germ cell transdifferentiation renders apoptosis resistant cells into apoptosis-sensitive cells (*Figure 1*), we hypothesized that it may allow the removal of cells from the tumorous gonad and thus may improve the health of the animals. To this end, we analyzed the progression of the germline tumor in *gld-1*-deficient animals in the presence or absence of ER stress. We found that *xbp-1* inactivation, not only induced germ cell transdifferentiation but also suppressed the germline tumor; demonstrating that the two phenomena are tightly correlated. The physiological improvement was manifested in several ways: (a) the density of the germ cells in the germline tumor was reduced in the tumorous animals exposed to ER stress such that the germline tumor that occupied the proximal gonad did not fill the entire gonad (*Figure 5A* and *Figure 6—figure supplement 1*). (b) as the germline tumor progresses, the gonad becomes over-packed with proliferating germ cells, resulting in increased rigidity of the gonad. Consequently, the motion of animals with a packed gonad germline tumor is limited resulting in paralyzed animals that can move their heads but cannot move their body. We find that ER stress significantly delayed the paralysis of the tumorous animals and allowed the animals to move freely at time-points where most of the non-stressed tumorous animals were paralyzed (*Figure 5B*, *Videos 1, 2* and *Figure 5—Figure supplement 1A*). This implies that the gonad of the animals subjected to ER stress were not as packed with germ cells as the non-stressed animals. (c) as the germline tumor progresses it ultimately kills the animal. We found that in wild-type animals, treatment with *gld-1* RNAi shortened the lifespan by more than 30%, whereas it shortened the lifespan of ER stressed-*xbp-1* mutants by only 13% (*Figure 5C*). Similarly, treatment with *gld-1* RNAi shortened wild-type animals lifespan by more than 30%, whereas it shortened the lifespan of *tfg-1* RNAi-treated animals by only 15% (*Figure 5—figure supplement 1B*). Importantly, no physiological improvement in terms of lifespan, movement or germline density were observed in *ire-1* deficient animals, in which ER stress-induced germ cell transdifferentiation does not occur, even upon *xbp-1* inactivation (*Figure 5* and *Figure 5—figure supplement 1*). Thus, the genetic requirements for suppression of the germline tumor and for induction of germ cell transdifferentiation converge on *ire-1*.

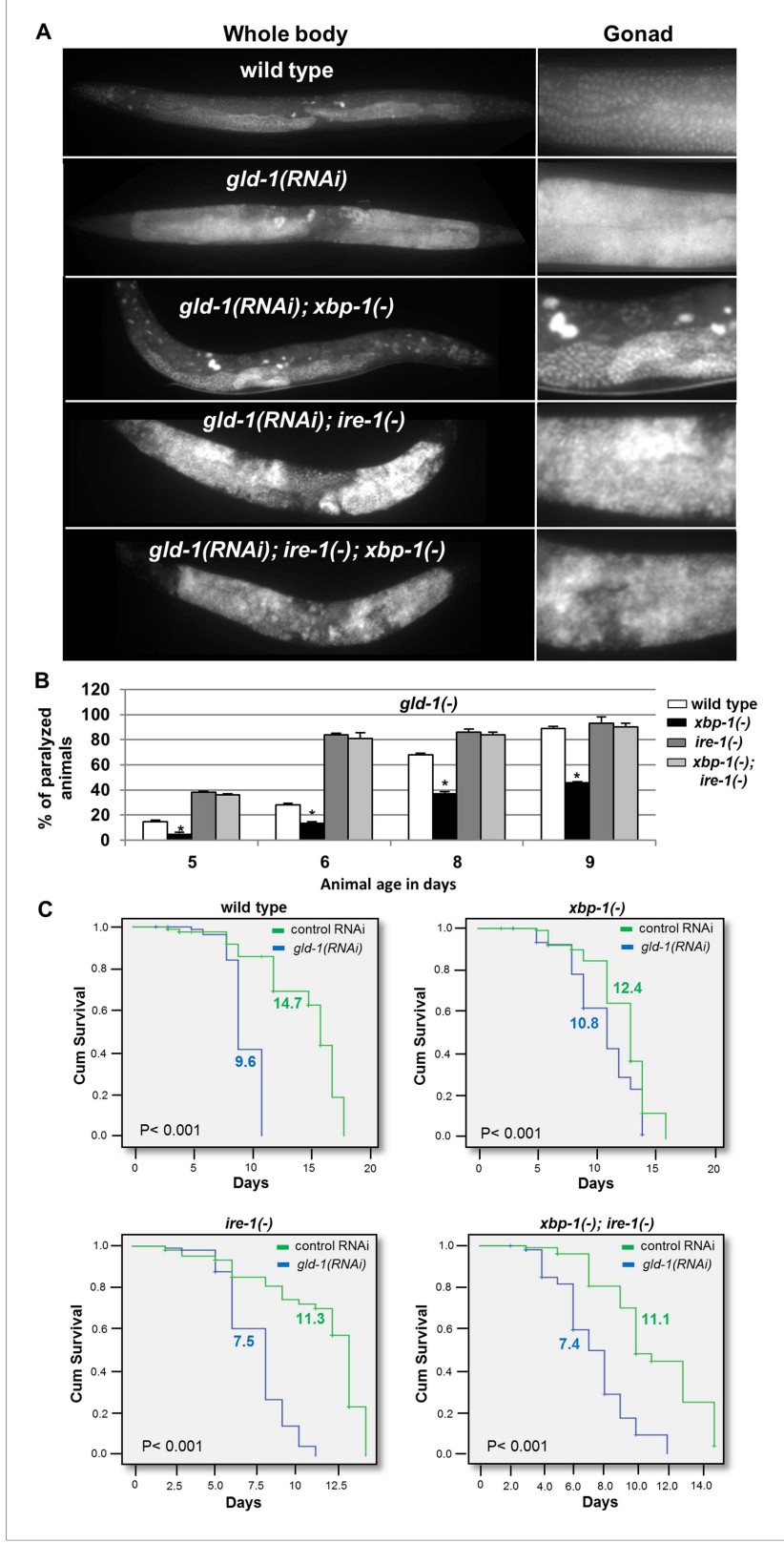

**Figure 5**. ER stress suppresses the germline tumor in an *ire-1*-dependent manner. (**A**) Representative micrographs showing whole body (x100) and gonads (x400) of day-4 animals stained with DAPI. (**B**) Paralysis assay in wild type, *xbp-1(tm2457)*, *ire-1(ok799)* and *xbp-1(tm2457); ire-1(ok799)* animals. At least 90 synchronized adult animals per genotype were placed on *gld-1*RNAi plates and their paralysis was scored on days 5, 6, 8 and 9. Bar graphs present
*Figure 5. continued on next page*

*Figure 5. Continued*

percentage of paralyzed animals. At all timepoints the *xbp-1* mutation significantly decreased the paralysis of the tumorous animals in an *ire-1*-dependent manner. Asterisks mark Student's T-test of p < 0.001 for reduced paralysis relative to wild-type animals. (**C**) Lifespan analysis of wild type, *xbp-1(tm2457)*, *ire-1(ok799)* and *xbp-1(tm2457); ire-1 (ok799)* animals treated with either *gld-1(RNAi)* to induce tumor formation or with control RNAi. Lifespan shortening was significantly suppressed in *xbp-1* mutants in an *ire-1*-dependent manner. Mean lifespan and p-values are indicated within each graph. *tfg-1* RNAi similarly supressed *gld-1*-RNAi-induced paralysis and lifespan shortening (see *Figure 5—figure supplement 1*).

The following figure supplement is available for figure 5:

**Figure supplement 1**. *tfg-1* RNAi suppresses the germline tumor in an *ire-1*-dependent manner.

## Not all stresses that induce germ cell apoptosis in non-tumorous gonads induce germ cell transdifferentiation in *gld-1*-deficient animals

DNA damage, pathogens, oxidative, osmotic, heat shock, starvation and ER stress are all known inducers of germ cell apoptosis in normal (i.e., non-tumorous) germlines (*Gartner et al., 2000*; *Salinas et al., 2006*; *Levi-Ferber et al., 2014*). We wondered whether such stresses that induce germ cell apoptosis in non-tumorous gonads would induce germ cell transdifferentiation in *gld-1*-deficient animals, similarly to ER stress. To address this, wild-type or *gld-1*-deficient animals were exposed to ER stress, genotoxic stress, mitochondrial stress or osmotic stress. These stresses were induced by manipulating genes whose inactivation is known to induce only one specific stress response, without globally stressing the animals. *tfg-1* RNAi was used to activate the ER stress response (*Levi-Ferber et al., 2014*). *rad-51* RNAi, which targets a DNA recombinase required for the repair of dsDNA breaks was used for the induction of genotoxic stress (*Gartner et al., 2000*). Mitochondrial stress was induced by RNAi targeting *ddl-3*, which encodes a kinesin light chain whose inactivation specifically activates the mitochondrial stress response (*Shore et al., 2012*). Osmotic stress was induced by inactivation of a negative regulator of the osmotic stress response, *osm-8* (*Rohlfing et al., 2011*). We confirmed that all of these treatments significantly increased the amount of apoptotic corpses in the gonads of animals with a non-tumorous germline (*Figure 6A* and white bar graph). We found that although some treatments induced more germ cell apoptosis than *tfg-1* RNAi, only treatment with *tfg-1* RNAi led to the detection of cells with large nuclei in the gonads of *gld-1; ced-3*-deficient animals (*Figure 6B* and black bar graphs). Thus, although ER stress induces germ cell apoptosis in normal gonads and hinders germ cell pluripotency in *gld-1*-deficient animals, the latter does not occur upon exposure to other severe cellular stresses, including genotoxic stress. This indicates that the transdifferentiation of the germ cells is not simply a side-effect or a default fate of apoptosis-resistant tumorous cells under proapoptotic stress which they fail to execute. Moreover, since only treatment with *tfg-1* RNAi suppressed the progression of the germline tumor (*Figure 6—figure supplement 1*), this lends further support to the notion that induction of germ cell transdifferentiation is a prerequisite for the suppression of the germline tumor.

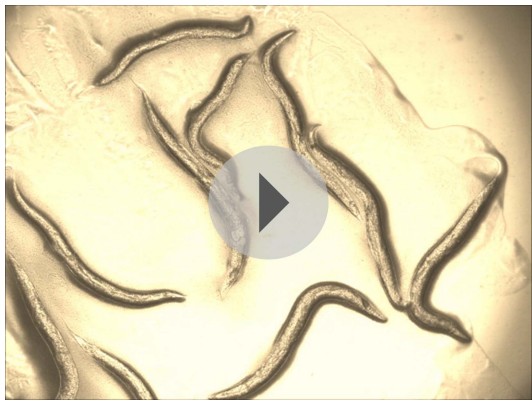

**Video 1.** Impaired motility of tumorous *gld-1(q485)* animals on day-4 of adulthood.

## Discussion

Understanding the molecular events that regulate germ cell fate in a normal germline, and even more so in a tumorous germline, is of fundamental importance in reproductive biology, stem cell research, and tumorigenesis and cancer therapy. In this study, we gained new and fascinating insights into the complex coupling between ER stress and germ cell fate in a *C. elegans* germline

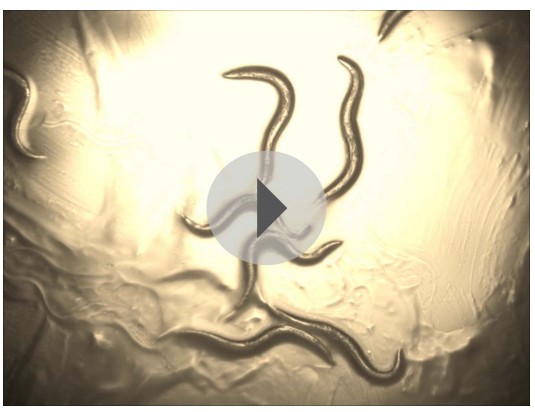

**Video 2.** ER stress induced by *tfg-1* RNAi treatment improved the motility of tumorous *gld-1(q485)* animals on day 4 of adulthood.

tumor model. We find that while severe stress limits organismal survival and well-being under most circumstances, ER stress is beneficial for the animals' healthspan in *gld-1*-deficient animals, as it suppresses their lethal germline tumors. This is in accordance with previous findings that a stress-inducing interference such as knockout of hsf1 in mice, which has deleterious impact on organismal survival under normal growth conditions as well as under stress, actually benefits the same organism in the case of cancer (*Dai et al., 2007*).

How does ER stress suppress the tumor in the *gld-1*-germline tumor model? Our findings indicate that the suppression of the germline tumor is the combined effect of germ cell transdifferentiation coupled with a limited half-life of the transdifferentiated germ cells (see model in *Figure 7*). The latter is achieved by restoring the responsiveness of the tumorous cells to apoptosis once they transdifferentiate into ectopic somatic cells and the subsequent removal of their ectopic somatic corpses from the gonad.

Importantly, although germ cell transdifferentiation normally occurs in *gld-1*-deficient animals (*Ciosk et al., 2006*), its extent is not sufficient to effectively suppress the tumor by itself. Effective suppression is only achieved under conditions that further enhance germ cell transdifferentiation beyond its basal level, as in the case of ER stress. All in all, our results indicate that tumor cell transdifferentiation may serve as a protective anti-tumor mechanism. This implies that transdifferentiation-promoting genes may effectively suppress tumors.

Many cellular stresses that induce germ cell apoptosis in normal gonads failed to affect germ cell pluripotency and tumor progression in *gld-1*-deficient animals. This suggests that the regulation of these processes in *gld-1*-deficient animals is specifically associated with ER stress, and is not simply a side-effect of apoptosis-resistant cells under stress. How might ER stress promote germ cell transdifferentiation? A major consequence of ER stress is interference with ER functions, which may directly impinge upon the production of a variety of signaling molecules such as secreted peptides, hormones, cholesterol and lipids which are metabolized in the ER. However, it is unlikely that this directly accounts for ER stress-induced germline transdifferentiation and apoptosis of the germ cells as these do not occur in animals under severe ER stress in the absence of *ire-1*. This suggests that it is the activation of the ER stress sensor IRE-1, rather than a non-specific consequence of ER stress per se, that regulate germ cell pro-differentiation and tumor progression in *gld-1*-deficient animals. Furthermore, since germ-cell transdifferentiation is promoted in animal deficient in *xbp-1*, this indicates that it is mediated by an *ire-1*-dependent *xbp-1* independent signal. What are the signaling molecules produced? Is it a single molecule or an arsenal of signaling molecules that regulate germ cell fate? These are all open questions to be addressed in future studies. Interestingly, in humans, mutations that compromise the activity or the expression of RNase L, an IRE1-related endoribonuclease, have been implicated with increased susceptibility to prostate cancer (*Carpten et al., 2002*; *Casey et al., 2002*). Unlike IRE1, RNase L has lost specificity to *xbp-1* mRNA. Nevertheless, it shares with IRE-1 its promiscuous RNase activity. This similarity may be sufficient as the tumor-suppressive properties of *ire-1* described herein are independent of *xbp-1*. This implies that tumor-suppressive properties of IRE-1-related endoribonucleases may be evolutionarily conserved, and may not be specifically associated with one specific type of prostate cancer.

One hallmark of aggressive tumors is their adaptation to their natural primary niche (*Hanahan and Weinberg, 2000*, *2011*). Thus, transformation into an ectopic kind of cell, rather than differentiation into a type of cell which can normally be found in the same primary niche, may have an added value in suppressing aggressive tumors. Strikingly, in 95% of human ovarian germline tumors, the totipotent germ cells precociously differentiate into a variety of ectopic somatic cells generating teratoma (*Koonings et al., 1989*; *Ulbright, 2005*). These naturally occurring teratoma are usually benign whereas the remaining 5% are usually malignant (*Ulbright, 2005*). This suggests that teratoma may be

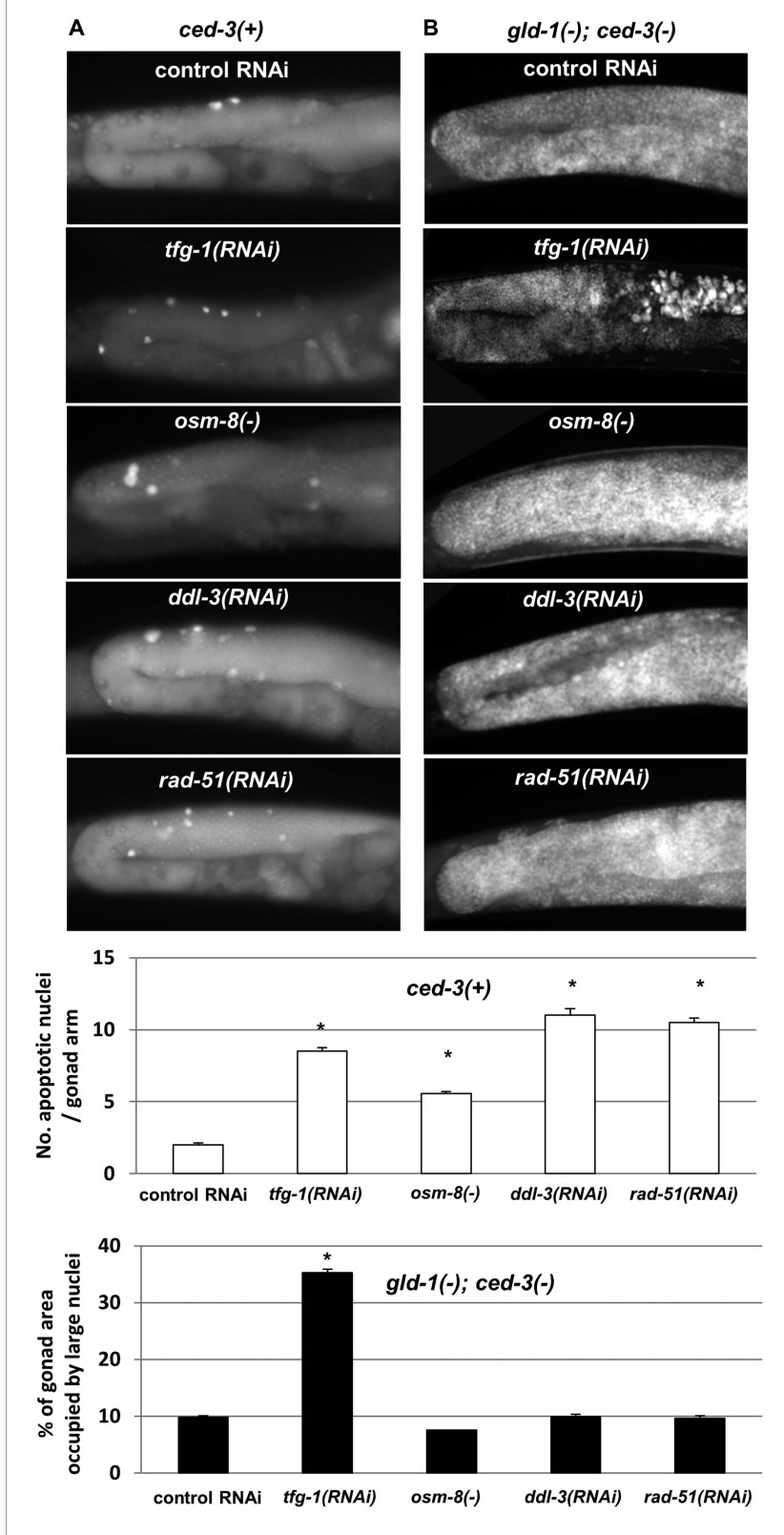

**Figure 6.** Loss of germ cell pluripotency in *gld-1*-deficient animals does not occur in response to all stresses. (**A**) Representative micrographs (x400) of gonads of *ced-3(+)* day 2 animals which were stained with SYTO12 to detect apoptotic cell corpses. White bar graphs present number of apoptotic nuclei per gonad arm (n = 60 gonads per genotype). (**B**) Representative micrographs (x400) of gonads of *gld-1(RNAi); ced-3(RNAi)* day-4 animals which *Figure 6. continued on next page*

*Figure 6. Continued*

were stained with DAPI to detect germline and somatic nuclei within the animals' gonads. Black bar graph presents the percentage of gonad area occupied by large nuclei (n = 60 gonads per genotype). ER stress was induced by *tfg-1* RNAi. Osmotic stress was induced by *osm-8* inactivation. Mitochondrial stress was induced by RNAi targeting *ddl-3*. Genotoxic stress was induced by *rad-51* RNAi. Asterisks mark Student's T-test of p < 0.001 relative to control RNAi-treated animals. Note that the stresses that did not induce germ cell transdifferentiation in *gld-1(–); ced-3(–)* animals also failed to suppress the germline tumor in *gld-1(–); ced-3(+)* animals (see **Figure 6—figure supplement 1**).

The following figure supplement is available for figure 6:

**Figure supplement 1**. Not all stresses suppress the germline tumor in *gld-1*-deficient animals.

a preferential germline tumor in terms of survival and fitness of the organism. As in other tumor-suppressive mechanisms, cancer cells may evolve to bypass this line of defense. Accordingly, some of the ovarian teratoma undergo a malignant transformation which occurs after the development of the teratoma. Nevertheless, this occurs only in 1.5% of teratoma (*Comerci et al., 1994*; *Ayhan et al., 2000*; *Ulbright, 2005*). This suggests that the conclusions derived from our studies in the *C. elegans* tumor model system are already naturally applied for human germline tumor biology, as if they were selected to do so by evolution.

The potential to suppress tumors by forcing their transdifferentiation requires the tumor cells to have differentiation potential. This raises the question whether transdifferentiation-mediated tumor suppression is relevant only for germline tumors, or is it relevant to a wider range of cell types? In mammals, in addition to the totipotent germ cells, subpopulations of stem cells, which maintain differentiation potential (i.e., pluri or multi-potent) exist in many organs and tissues, albeit at low numbers (*Reya et al., 2001*). Furthermore, within tumors and hematological cancers, one can find a subpopulation of cancer stem cells that possess stem cells characteristics, including the ability to give rise to a variety of cell types (*Reya et al., 2001*). Accordingly, differentiation therapy has been used in the clinic and is proven to be effective in limiting some kinds of tumors (*Sell, 2004*). Interestingly, whereas these treatments induce differentiation of cells within the same lineage, induction of transdifferentiation into a completely different cellular lineage may be even more effective by creating a discrepancy and maladaptiveness between the tumorous cells and their surrounding niche. Thus, induction of tumor cell transdifferentiation has the potential to be applied as a novel approach to combat cancer and overcome the escape of tumor cells from the cell death machinery in a wide variety of tissues.

## Materials and methods

### Cell corpse assays

The number of apoptotic cells in the gonads of day-2 or day-3 animals was assessed by scoring the number of SYTO12/CED-1::GFP labeled cells in the gonad. SYTO12 (Molecular Probes, Eugene, Oregon) staining was performed as previously described (*Gumienny et al., 1999*).

### Stress treatment

Day-0 animals were placed on plates containing 45 µg/ml tunicamycin (Calbiochem, Billerica, MA). SYTO12 staining and expression of somatic transgenes were analyzed on day 3 or day 4 of adulthood.

### RNA interference

Bacteria expressing dsRNA were cultured overnight in LB containing tetracycline and ampicillin. Bacteria were seeded on NGM plates containing IPTG and carbenicillin. RNAi clone identity was verified by sequencing. Eggs were placed on plates and synchronized from day-0 (L4). The efficacy of the *tfg-1* RNAi was confirmed by the animals' reduced body size (*Witte et al., 2011*). The efficacy of the *ced-3* RNAi was confirmed by the lack of apoptotic corpses in the gonads. The efficacy of the *gld-1* RNAi was confirmed by the tumorigenicity of the gonads, by the absence of oocytes and embryos and by western blotting. Some experiments involved double or triple RNAi mixtures, in which the relative

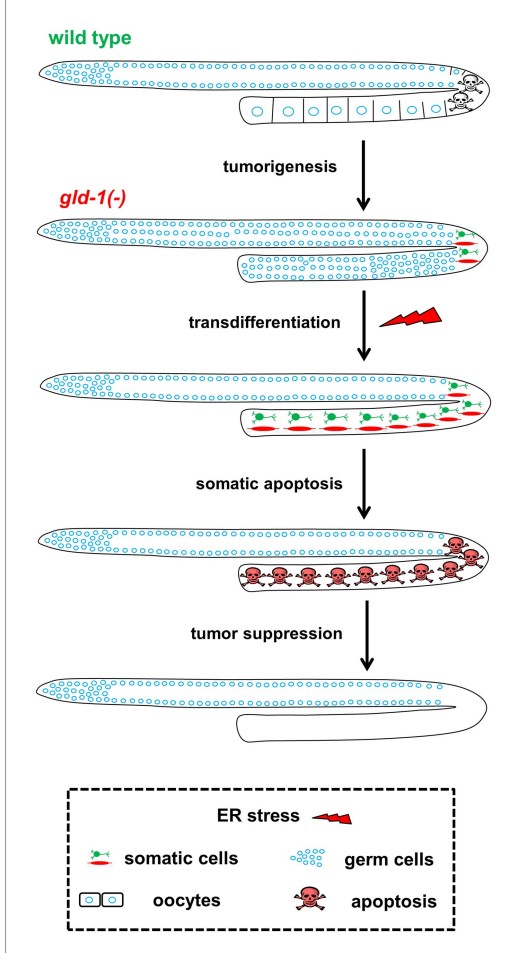

**Figure 7**. Model—tumor progression and germline fate in ER stressed tumorous gonads. In animals with a normal germline, ER stress induces germ cell apoptosis. Inactivation of the *gld-1* genes alters many aspects of germ cell fate: differentiation of germ cells into oocytes is abrogated, germ cell proliferation is enhanced, a germline tumor is formed and the germ cells lose their responsiveness to execute physiological and stress-induced apoptosis. Furthermore, in *gld-1* deficient animals the germ cells are prone to generate teratoma as they become sensitized to precociously transdifferentiate into somatic cells. Under these conditions ER stress can suppress and limit the germline tumor. This suppression is achieved by enhancing germline transdifferentiation into ectopic somatic cells. Soon after the transdifferentiation, these ectopic cells undergo apoptosis, and are removed from the gonad, suppressing the germline tumor.

amount of each RNAi bacteria was kept equal between samples by supplementing with control RNAi as needed.

## Fluorescence microscopy and quantification

To follow expression of fluorescent transgenic markers, transgenic animals were anaesthetized on 2% agarose pads containing 2 mM levamisol. Images were taken with a CCD digital camera using a Nikon 90i fluorescence microscope. For each trial, exposure time was calibrated to minimize the number of saturated pixels and was kept constant through the experiment. The NIS element software was used to quantify mean fluorescence intensity as measured by intensity of each pixel in the selected area within the gonad. To determine the fraction of the gonad area occupied by ectopic cells day-4 animals were fixed and stained with DAPI. The NIS element software was used to manually select and quantify the gonad area as well as the area within the gonad that was occupied by abnormal DAPI-stained nuclei in the tumorous animals.

## Western blot

A similar number of animals were boiled in protein sample buffer containing 2% SDS. Proteins were separated using standard PAGE separation, transferred to a nitrocellulose membrane and detected by western-blotting using anti-GLD-1 (1:1000, kindly provided by Prof Anton Gartner [*Rutkowski et al., 2011*]) and anti-tubulin (DHSB, 1:5000).

## Lifespan and paralysis assay

RNAi treatments were performed continuously from the time of hatching. Eggs were placed on plates seeded with the RNAi bacteria of interest. Paralysis and lifespan were scored every 1–2 days. Related lifespans were performed concurrently to minimize variability. In all experiments, lifespan was scored as of the L4 stage which was set as t = 0. Animals that ruptured or crawled off the plates were included in the lifespan analysis as censored worms. SPSS program was used to determine the means and the p values. p values were calculated using the log-rank (i.e., Mantel–Cox) method.

## Statistical analysis

Error bars represent the standard error of the mean (SEM) of at least 3 independent experiments. Except for lifespan analysis, p values were calculated using the unpaired Student's t test.

## Strains and transgenic lines

The following lines were used in this study: N2, CF2012: *pek-1(ok275) X*, CF2988: *atf-6(ok551) X*, CF2473: *ire-1(ok799) II*, CF3208: *xbp-1(tm2457) III*, CF2472: *xbp-1(zc12) III*, SHK62: *ire-1(ok799) II*;

*xbp-1(tm2457) III*, MD701: *Plim-7:: ced-1::gfp V*, CF2185: *ced-3(n1286) IV*, DP132: *edIs6 (punc-119:: GFP) IV*, SHK75: *irIS25 (pJM86; pelt-2::NLS:GFP::LacZ + rol-6) V*, SHK40: *glp-1(e2141) III*; irIS25 *(pJM86; pelt-2::NLS::GFP::LacZ + rol-6) V*, MT3571: *osm-8(n1518) II*, SHK118: *gld-1(q485)/unc-13 (e51) I*; *ced-3(n1286) IV*, SHK152: *ced-3(n1286) edIs6 (Punc-119::gfp) IV*.

## Acknowledgements

We thank Prof Jeremy Don (Bar-Ilan University, Israel) and members of the Korenblit lab for helpful discussions. We thank Prof Anton Gartner (College of Life Sciences, University of Dundee, Dundee) for the GLD-1 antibodies. We thank Dr Shohei Mitani (National Bioresource Project for the nematode, Tokyo Women's Medical University School of Medicine, Japan) and the *Caenorhabditis* Genetics Center for providing nematode strains. All relevant data are within the paper and its Supporting Information files.

## Additional information

### Funding

| Funder | Grant reference | Author |
| --- | --- | --- |
| Israel Science Foundation | Israel Science Foundation 1749/11 to SHK | Sivan Henis-Korenblit |

The funder had no role in study design, data collection and interpretation, or the decision to submit the work for publication.

### Author contributions

ML-F, Conception and design, Acquisition of data, Analysis and interpretation of data, Drafting or revising the article; HG, RD, Conception and design, Acquisition of data, Analysis and interpretation of data; SH-K, Conception and design, Analysis and interpretation of data, Drafting or revising the article

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
