## [Decision Letter]

Your paper entitled “Transdifferentiation mediated tumor suppression by endoplasmic reticulum stress in *C. elegans*” has been independently reviewed by two experts and a member of *eLife's* Board of Reviewing Editors. The review process was followed by consultations between these three that led to consensus presented below.

The importance of your paper's subject matter and the significance of your discovery that manipulating ER stress and the response to it affects differentiation and tumor formation in the *gld-1* deficient *C. elegans* model were recognized by all three reviewers. However, the review process unearthed certain limitations pertaining to the strength of your conclusions that need to be addressed prior to publication in *eLife*.

The Abstract and Title claims that ER stress suppresses tumor formation, however the experiments displayed merely show that elimination of *xbp-1* suppresses tumor formation in *gld-1* deficient worms. To support the claim made in the Abstract, it would be important to present evidence that manipulations that lead to ER stress indeed suppress tumor formation (and not merely lead to trans differentiation, as is presently shown). The interest of the observations would not be diminished were it to turn out that elimination of *xbp-1* is a pre-requisite to tumor suppression; for example if compromising *tfg-1* by RNAi further suppresses tumors of *gld-1* deficient worms in an *ire-1* dependent manner only when *xbp-1* is also missing, but the Title, Abstract and Discussion would need to accommodate such a finding.

The Ploegh and Glimcher labs have shown that elimination of XBP1 in mouse leads to higher levels of IRE-1 protein and enhanced phosphorylation of this UPR effector (see Figure 2 in PMID 19407814 and Figure 3 in PMID: 22291093). Your case for an *ire-1* mediated tumor suppression mechanism in worms would be strengthened by evidence that something similar is going on in the *xbp-1* deficient worms; if this is experimentally unfeasible it remains an important discussion point. RNaseL, an IRE-1 descendent (evolutionarily-speaking), which has retained its promiscuous RNase activity but has lost specificity to XBP1, has been implicated in tumor suppression in mammals (PMID: 11799394 and 12415269). These facts stand to inform the discussion of your findings. More broadly, the three reviewers of your paper feel that it might be improved if its thrust were to be shifted from the role of ER unfolded protein stress in trans differentiation and tumor suppression to a role for *ire-1* (hyper) activity in these processes.

The emergence of cells with large nuclei and the acquisition of differentiation markers in stressed/*xbp-1* deficient worms are correlated phenomena. An attempt should be made to establish if these processes affect the same cells (do the cells expressing the differentiation markers have larger nuclei?) and the outcome of this attempt presented and discussed.

In this vein, *ire-1* dependent trans differentiation and tumor suppression are correlated phenomena but a causal link between them has not been established in your study. This point should be emphasized for reader edification.

The above consensus statement should serve as the basis for revision of your paper. However in the interest of complete transparency, in this case the unedited original reviews are provided below.

*Reviewer #1*:

The key observation made here as non-trivial: The authors claim to have discovered that pharmacological (tunicamycin) and genetic (*tfg-1* RNAi) manipulations that impede protein folding homeostasis in the ER lead to trans-differentiation of “tumorous” germ cells in *gld-1* deficient worms. Left there, this would subject to many different interpretations. But what makes their findings potentially interesting is the observation that such stress-induced trans-differentiation is dependent on IRE-1 signaling but is independent of IRE-1's main known effector XBP1. In fact, inactivation of XBP1 on its own promotes such salutary trans-differentiation of the “tumorous” germ cells in *gld-1* deficient worms. These observations argue against trans-differentiation reflecting a non-specific feature of mounting levels of ER stress and rather suggest that some aspect of IRE-1 signaling that is independent of XBP1 sets up this anti-tumour defense mechanism. The echoes to RNaseL, which is a derivative of IRE-1 with tumour suppressive properties, are interesting.

The story more or less ends there, but this may be enough of a new insight into oncogenesis to merit consideration as brief report.

There are obstacles to this path: The authors feel they have made not one but three important discoveries (as per the cover letter) and they are apparently not aware of the RNaseL cancer connection, which adds important context to what is essentially a one-finding paper.

Thus the question before the three reviewers is whether this is a sufficient advance and if it can be packaged succinctly to deliver a story worthy of *eLife*.

*Reviewer #2*:

Tumor cells have acquired resistance to apoptosis that limits the effectiveness of many therapeutic strategies. In this original study, the authors demonstrate in a *C. elegans* model how tumor growth can be suppressed by changing tumor cell identity. Prior studies of others showed that deficiency in *gld-1*, encoding a translational suppressor, results in proximal germline tumors in the gonad, and that some of the tumorous germ cells undergo trans-differentiation into ectopic somatic cells. The authors show that activation of IRE-1 by ER stress inhibited progression of a lethal *gld-1* deficient germline tumor through trans-differentiation. Restoration of the ability to induce apoptosis accompanied the cell transition. It is suggested that tumor cell trans-differentiation will counteract the escape of tumor cells from apoptotic cell death with possible therapeutic benefits.

Results show:

Figure 1 and Figure 1—figure supplement 1 and Figure 1—figure supplement 2. ER stress induced in *gld-1* deficient animals by either treatment with tunicamycin or down-regulating *tfg-1* resulted in apoptotic cell corpses in the tumorous gonads. Results were confirmed with CED-1:GFP expression in animals stressed with *tfg-1* RNAi. The apoptotic cell corpses differed in size, shape and location in tumorous as compared with non-tumorous gonads.

Figure 2. Blocking apoptosis by depleting or preventing *ced-3* expression caused ectopic cells with large nuclei to accumulate in *gld-1* depleted ER stressed animals.

Figure 3. The cells with large nuclei in the *gld-1*-depleted apoptosis-defective animals are germ cell-derived somatic cells as determined with differentiation markers.

Figure 4. Interestingly, trans-differentiation in response to *tfg-1* RNAi was blocked in *ire-1* deficient animals. The conclusion is that trans-differentiation in response to ER stress requires IRE-1. But furthermore, the ER-stress induced trans-differentiation was independent of its splicing target mRNA for the transcription factor, *xbp-1*. The implications are that the IRE-1 mechanism involves other signaling events (besides *xbp-1* splicing) and/or the RIDD pathway involving mRNA decay. [It might be worth mentioning that in humans the IRE-1-related endoribonuclease, RNase L, has been suggested in several studies to have tumor suppressing activities (e.g. PMID: 11799394 and 12415269)].

Figure 5. ER stress inhibited tumor growth, delaying paralysis and increasing lifespans, but not in IRE-1 deficient animals.

Figure 6. Deficiency in genes that cause different types of cellular stress revealed only *tfg-1* depletion which induces ER stress led to trans-differentiation in *gld-1* deficient animals and suppresses germline tumors (Figure 6—figure supplement 1).

The concept that trans-differentiation of tumor cells into somatic cells can prevent tumor progress by rendering cells susceptible to ER stress induced apoptosis is well supported by this study. It is particularly interesting that IRE-1 dependent, but *xbp-1* independent signaling is necessary for regulation of germ cell fate. This study is provocative and could lead to many downstream investigations into mechanisms and into whether the general concepts presented here are applicable to cancer in humans.

*Reviewer #3*:

This paper describes an effect of the UPR on transdifferentiation and apoptosis in a worm germline tumor model. The main conclusions are: 1. Activation of Ire-1 induces germ cell transdifferentiation into ectopic somatic cells, 2. This effect leads to apoptosis of the ectopic cells (which would normally by resistant to apoptosis) and 3. This process suppresses the lethal effects of the tumor.

The evidence that the cells in question are transdifferentiated somatic cells is shown in Figure 3, where several somatic cell markers are shown to increase with ER stress. The authors conclude that “the cells with abnormal large nuclei, which are detected in apoptosis-defective *gld-1* RNAi-treated animals, and which are further induced by ER stress, appear to be of somatic nature” (subsection headed “The ectopic cells in the gonad of *gld-1*-deficient animals are germ cell-derived somatic cells”). I would have liked to see more evidence that these are indeed the same cells. The units in Figure 3 are relative fluorescence rather than % gonad filled like the other figures, so it is difficult to discern whether these transdifferentiated cells can account for the all of the large-nuclei cells that accumulate in the tumor model when apoptosis is inhibited. Perhaps a double-labeling experiment with the somatic cell markers and dapi would clear this up.

The fact that the accumulation of the ectopic cells is dependent on Ire-1 but not XBP-1 is interesting. The authors discuss at several points that ER stress promotes transdifferentiation/tumor suppression etc., but in the Discussion suggest that it is not ER stress per se but Ire-1 signaling that causes this effect. This seems consistent with the data. The authors should point out that in *xbp1* mutants (at least in several tissue-specific examples in mice), IRE-1 is often over-activated. Does this occur in the *xbp-1* mutant worms as well? This could further explain why the *xbp1* mutants suppress the tumor lethality.

The tumor suppression referred to in the Title of the manuscript is actually shown only for *xbp1* mutants, and not in cases of induced ER stress. In the subsection headed “ER stress-induced germ cell transdifferentiation suppresses the germline tumor”, the authors state that inactivating *tfg-1* or *xbp-1* suppressed the germline tumor, but Figure 5 does not have any data for *tfg-1*. In order to support the claim made in the Title, I think the authors should either show tumor suppression in a more clear-cut case of ER stress (chemical or genetic), or they should show tumor suppression in a more clear-cut case of IRE-1 activation (e.g., overexpression of IRE) and change the Title to reflect this.

Finally, is anything known about the molecular pathways and genes that underlie transdifferentiation in this system? More discussion is warranted if so- could the authors propose possible targets of *ire-1* that could mediate this effect (either JNK signaling or RIDD)?

---

## [Author Response]

*The Abstract and Title claims that ER stress suppresses tumor formation, however the experiments displayed merely show that elimination of* xbp-1 *suppresses tumor formation in* gld-1 *deficient worms. To support the claim made in the Abstract, it would be important to present evidence that manipulations that lead to ER stress indeed suppress tumor formation (and not merely lead to trans differentiation, as is presently shown). The interest of the observations would not be diminished were it to turn out that elimination of* xbp-1 *is a pre-requisite to tumor suppression; for example if compromising* tfg-1 *by RNAi further suppresses tumors of* gld-1 *deficient worms in an* ire-1 *dependent manner only when* xbp-1 *is also missing, but the Title, Abstract and Discussion would need to accommodate such a finding*.

We now show that similarly to ER stress induced by the *xbp-1* deficiency, ER stress induced by *tfg-1* RNAi also suppresses the germline tumor in *gld-1*-deficient animals as can be seen by the improvement in the physiology of the RNAi-treated animals in terms of paralysis, in terms of lifespan and as reflected by the reduced density of germ cells in the gonads (see Figure 6—figure supplement 1, Figure 5—figure supplement 1 and subsection headed “ER stress suppresses the germline tumor in an *ire-1* dependent manner”).

*The Ploegh and Glimcher labs have shown that elimination of XBP1 in mouse leads to higher levels of IRE-1 protein and enhanced phosphorylation of this UPR effector (see*
Figure 2
*in PMID 19407814 and*
Figure 3
*in PMID: 22291093). Your case for an* ire-1 *mediated tumor suppression mechanism in worms would be strengthened by evidence that something similar is going on in the* xbp-1 *deficient worms; if this is experimentally unfeasible it remains an important discussion point*.

The best available tool in *C. elegans* to follow IRE-1 activation is to monitor *xbp-1* splicing. This is of course not possible in animals whose *xbp-1* gene is deleted (as in the *xbp-1(tm2457)* allele that has been used in this study. However, in an *xbp-1(zc12)* mutant, which contains an early nonsense mutation at codon 34, the *xbp-1* splicing site remains intact. In the past we have demonstrated that the level of *xbp-1* splicing in *xbp-1(zc12)* mutants was significantly higher than in wild-type animals, indicating that *xbp-1* deficiency leads to IRE-1 activation in *C. elegans*, as it does in mammals (Safra et al., JCS 2013). Presumably, this is also the case in the *xbp-1(tm2457)* null mutants.

Since this study made use of the *xbp-1(tm2457*) null mutants and not the *xbp-1(zc12)* mutants, we used DAPI staining to examine whether abnormally large nuclei accumulated in the gonads of *gld-1*-deficient animals carrying the *xbp-1(zc12)* mutation. We found that the *xbp-1(zc12)* mutation increased germ cell transdifferentiation similarly to the *xbp-1(tm2457)* mutation (P=0.23) (see Figure 4 and the second paragraph of the subsection headed “*ire-1* promotes ER stress-induced germ cell transdifferentiation in the gonad of *gld-1-*deficient animals independently of *xbp-1*”).

*RNaseL, an IRE-1 descendent (evolutionarily-speaking), which has retained its promiscuous RNase activity but has lost specificity to XBP1, has been implicated in tumor suppression in mammals (PMID: 11799394 and 12415269). These facts stand to inform the discussion of your findings. More broadly, the three reviewers of your paper feel that it might be improved if its thrust were to be shifted from the role of ER unfolded protein stress in trans differentiation and tumor suppression to a role for* ire-1 *(hyper) activity in these processes*.

We now discuss the established activity of the RNaseL in tumor suppression and emphasize that in our settings, ER stress induced germ cell transdifferentiation and suppresses the germline tumor, not directly due to the ER stress but rather due to its ability to activate IRE-1 (see Discussion).

*The emergence of cells with large nuclei and the acquisition of differentiation markers in stressed/*xbp-1 *deficient worms are correlated phenomena. An attempt should be made to establish if these processes affect the same cells (do the cells expressing the differentiation markers have larger nuclei?) and the outcome of this attempt presented and discussed*.

Two independent lines of experiment suggest that indeed the cells expressing the somatic markers and the cells with the large nuclei are the same cells:

1) In the case of the intestinal marker, the reporter gene contained an NLS signal and the labeled nuclei were indeed larger than typical germ cells (see Figure 3).

2) By combining Hoecst-nuclei staining and a strain expressing the neuronal marker *Punc-119::gfp*, we now show that the cells expressing the neuronal differentiation marker co-localize with the large nuclei in the gonad (Note that only some of the cells with the large nuclei also express the neuronal marker, consistent with the fact that the teratoma contains somatic cells of the different germ-layers; see Figure 3—figure supplement 1).

In addition, in a third experiment, by generating a strain that co-expresses c*ed-1::gfp* in the sheath cells of the gonad as well as the intestinal marker *Pelt-2::gfp::NLS*, we now show that cells expressing the somatic markers are being engulfed and cleared from the gonad via phagosomes. (Note that here too only some of the cells that are being engulfed away from the gonad express the neuronal marker, consistent with the fact that the teratoma contains somatic cells of the different germ-layers; see Figure 3—figure supplement 1).

These experiments are presented in the subsection headed “The ectopic cells in the gonad of *gld-1-*deficient animals express somatic markers”.

*In this vein,* ire-1 *dependent trans differentiation and tumor suppression are correlated phenomena but a causal link between them has not been established in your study. This point should be emphasized for reader edification*.

We are now more careful with this statement, and corrected it in the Abstract, in the Results section and in the Discussion.

Reviewer #2:

*[…]*
Figure 4*. Interestingly, trans-differentiation in response to* tfg-1 *RNAi was blocked in* ire-1 *deficient animals. The conclusion is that trans-differentiation in response to ER stress requires IRE-1. But furthermore, the ER-stress induced trans-differentiation was independent of its splicing target mRNA for the transcription factor,* xbp-1*. The implications are that the IRE-1 mechanism involves other signaling events (besides* xbp-1 *splicing) and/or the RIDD pathway involving mRNA decay. [It might be worth mentioning that in humans the IRE-1-related endoribonuclease, RNase L, has been suggested in several studies to have tumor suppressing activities (e.g. PMID: 11799394 and 12415269)]*.

RNaseL role in tumor suppression is now discussed in the Discussion section.

Reviewer #3:

*[…] The evidence that the cells in question are transdifferentiated somatic cells is shown in*
Figure 3*, where several somatic cell markers are shown to increase with ER stress. The authors conclude that* “*the cells with abnormal large nuclei, which are detected in apoptosis-defective* gld-1 *RNAi-treated animals, and which are further induced by ER stress, appear to be of somatic nature*” *(subsection headed “The ectopic cells in the gonad of* gld-1*-deficient animals are germ cell-derived somatic cells”). I would have liked to see more evidence that these are indeed the same cells. The units in*
Figure 3
*are relative fluorescence rather than % gonad filled like the other figures, so it is difficult to discern whether these transdifferentiated cells can account for the all of the large-nuclei cells that accumulate in the tumor model when apoptosis is inhibited. Perhaps a double-labeling experiment with the somatic cell markers and dapi would clear this up*.

A double-labeling experiment with the somatic cell markers and hoecst staining has been done and should clear co-localization between cells expressing the somatic marker and cells with large nuclei within the gonad. (Figure 3—figure supplement 1).

*The fact that the accumulation of the ectopic cells is dependent on* IRE-1 *but not* XBP1 *is interesting. The authors discuss at several points that ER stress promotes transdifferentiation/tumor suppression etc., but in the Discussion suggest that it is not ER stress per se but* IRE-1 *signaling that causes this effect. This seems consistent with the data. The authors should point out that in* xbp1 *mutants (at least in several tissue-specific examples in mice),* IRE-1 *is often over-activated. Does this occur in the* xbp-1 *mutant worms as well? This could further explain why the* xbp1 *mutants suppress the tumor lethality*.

In the past we have demonstrated that the level of *xbp-1* splicing in *xbp-1(zc12)* mutants was significantly higher than in wild-type animals, indicating that *xbp-1* deficiency leads to IRE-1 activation in *C. elegans*, as it does in mammals (Safra et al., JCS 2013).

*The tumor suppression referred to in the Title of the manuscript is actually shown only for* xbp1 *mutants, and not in cases of induced ER stress. In the subsection headed “ER stress-induced germ cell transdifferentiation suppresses the germline tumor*”*, the authors state that inactivating* tfg-1 *or* xbp-1 *suppressed the germline tumor, but*
Figure 5
*does not have any data for* tfg-1*. In order to support the claim made in the Title, I think the authors should either show tumor suppression in a more clear-cut case of ER stress (chemical or genetic), or they should show tumor suppression in a more clear-cut case of IRE-1 activation (e.g., overexpression of IRE) and change the Title to reflect this*.

We now show that similarly to ER stress induced by the *xbp-1* deficiency, ER stress induced by *tfg-1* RNAi also suppresses the germline tumor in *gld-1-*deficient animals in an *ire-1-*dependent manner. This can be seen by the improvement in the physiology of the RNAi-treated animals in terms of paralysis, in terms of lifespan and as reflected by the reduced density of germ cells in the gonads (Figure 5—figure supplement 1 and Figure 6—figure supplement 1).